# Using Triethylborane to Manipulate Reactivity Ratios in Epoxide-Anhydride Copolymerization: Application to the Synthesis of Polyethers with Degradable Ester Functions

**DOI:** 10.3390/molecules27020466

**Published:** 2022-01-11

**Authors:** Vamshi K. Chidara, Yves Gnanou, Xiaoshuang Feng

**Affiliations:** Physical Sciences and Engineering Division, King Abdullah University of Science and Technology (KAUST), Thuwal 23955-6900, Saudi Arabia; vamshi.chidara@kaust.edu.sa

**Keywords:** triethylborane, reactivity ratios, anionic polymerization, degradable polymers, poly(ethylene oxide)

## Abstract

The anionic ring-opening copolymerization (ROCOP) of epoxides, namely of ethylene oxide (EO), with anhydrides (AH) generally produces strictly alternating copolymers. With triethylborane (TEB)-assisted ROCOP of EO with AH, statistical copolymers of high molar mass including ether and ester units could be obtained. In the presence of TEB, the reactivity ratio of EO (*r*_EO_), which is normally equal to 0 in its absence, could be progressively raised to values lower than 1 or higher than 1. Conditions were even found to obtain *r*_EO_ equal or close to 1. Samples of P(EO-*co*-ester) with minimal compositional drift could be synthesized; upon basic degradation of their ester linkages, these samples afforded poly(ethylene oxide) (PEO) diol samples of narrow molar mass distribution. In other cases where *r*_EO_ were lower or higher than 1, the PEO diol samples eventually isolated after degradation exhibited a broader distribution of molar masses because of the compositional drift of initial P(EO-*co*-ester) samples.

## 1. Introduction

During copolymerization, comonomers are inserted in the chain at a rate and in a sequence order that is dictated by two factors that are the monomers’ reactivity ratios and the composition of the initial monomer mixture. The distribution of monomers along the chain thus varies in most copolymers from chain to chain. In contrast, in sequence-controlled copolymers, the two comonomers are arranged in a same precise order. Alternating copolymers, periodic copolymers, and block copolymers are examples of sequence-controlled copolymers that are obtained when reactivity ratios of both comonomers are either equal to or largely higher than one [1,2]. For instance, Frey’s group reported the anionic copolymerization of two different monomers characterized by a large difference in their reactivity ratios that resulted in the formation of block-like tapered copolymers [3,4]. Likewise, Wurm’s group reported the anionic copolymerization of aziridines and ethylene oxide (EO), which afforded blocky structures because of the large difference in the reactivity ratios of the two monomers [5]. William’s group achieved the control of polymer composition through the use of a catalyst that switched from the copolymerization of epoxides with various comonomers to the homopolymerization of ε-caprolactone, resulting in the formation of multiblock copolymers [6]. Frey’s group also reported that i-Bu_3_Al and double metal cyanide catalysts, when used to copolymerize ethylene oxide and propylene oxide, produce in the first case blocky structures and in the second statistical copolymers [7].

The above works thus resorted to different strategies to control monomer distributions along the polymer chain. The authors of these investigations selected monomers with either substantially large differences in their activities or specific catalyst systems to achieve control of monomer distribution. As shown in the above examples, EO is one such monomer that lends itself to all sorts of copolymerizations, including alternating, block, and random copolymerizations. Examples of comonomers that can be selectively incorporated into polymer chains as a function of their electrophilic nature as well as the nucleophilic nature of the propagating chain-end are numerous.

Let us now focus on the case of EO and of poly(ethylene oxide) (PEO) or poly(ethylene glycol) (PEG), a polymer used in particular for drug delivery applications because of its biocompatibility, hydrophilicity, chemical stability, and its stealth capability [8]. Commercial PEO/PEG polymers range from short oligomer chains to long polymer chains. For pharmaceutical grade, therapeutical applications, PEG should be less than 40 kDa molar mass to ensure efficient glomerular clearance [9]. One approach to synthesize PEG is to resort to the anionic polymerization of EO triggered by alkaline initiators (NaOH, KOH etc.). Another approach is to incorporate degradable moieties in the high-molar-mass PEO backbone and then to degrade into low-molar-mass PEG [10,11,12]. Our group actually reported the metal-free anionic ROCOP of EO with lactide that produced high-molar-mass PEO with few ester linkages that were subsequently degraded into low-molar-mass PEG [12]. Copolymerization systems that involve epoxides and CO_2_ or anhydrides have also been manipulated using TEB to generate copolymers with a random distribution of carbonate and ester functions [13,14,15]. Following our disclosure, Zhang and coworkers showed that in the presence of a smaller amount of TEB, the system automatically switches from the alternating copolymerization of EO with anhydrides to the homopolymerization of EO upon initially mixing the two monomers [16,17].

As mentioned above, epoxides and anhydrides tend normally to copolymerize in an alternate way when initiated by an anionic species. Taking advantage of the disruptive presence and role of TEB, we demonstrate in this work that in the presence of the latter Lewis acid EO and anhydrides copolymerize statistically, resulting in a random distribution of ester functions along the polyether chain formed. We demonstrate that the reactivity ratio of EO (*r*_EO_) in a regular EO/anhydride copolymerization that is equal to 0 can be manipulated by the amount of TEB used and increased well above 1 to generate various polymer modes (Figure 1). While there are various existing methods to determine reactivity ratios, we choose the nonterminal Jaacks model [18] to determine the values taken by the reactivity ratios of EO (*r*_EO_) in ROCOP of EO with anhydrides as a function of TEB content. As a practical application of this kinetic study, various samples of high molar mass PEO including randomly distributed ester functions were prepared and purposely degraded to generate low molar mass telechelics.

## 2. Results and Discussion

Our research group first reported the TEB-assisted ROCOP of epoxides with CO_2_ highlighting the importance of TEB to avoid the formation of cyclic carbonates and promote the growth of linear polycarbonates [13]. In a subsequent publication, we also demonstrated that TEB dramatically influences the terpolymerization of epoxides, anhydrides and CO_2_ [14]. Depending upon the feeding ratio of anhydrides to epoxides, either tapered block poly(ester-*b*-carbonate) copolymers or random poly(ester-*co*-carbonate) copolymers were eventually obtained.

In this investigation, which is related to the copolymerization of EO with anhydrides and to the influence of TEB, we first carried out a blank reaction by mixing 100 eq. of EO with 5 eq. of succinic anhydride (SA), in absence of TEB using 1 eq. of tetrabutylammonium chloride (TBACl) as initiator. Prior to this reaction, we checked that no reaction occurred when TBACl was not added to the reaction mixture, confirming that TEB by itself cannot initiate copolymerization (entry 1, Table 1). Only 11% SA and <1% EO was converted to poly(ethylene succinate) in absence of any TEB (entry 2, Table 1); similar results were observed for other epoxides and anhydrides by other groups [19,20,21]. It thus appears from these two reactions that both TBACl as an initiator and TEB as an activator are crucial for the successful ROCOP of EO and SA. As mentioned in our prior work and as observed by other groups, the presence of TEB in the reaction mixture mainly play a dual role [13,14,19,20].

a. TEB forms an ate complex with propagating oxyanion of epoxide or anhydride, which is weak enough not to indulge in side reactions with the ester functions carried by the chain,

b. the excess of TEB used helps to activate both epoxide or anhydrides facilitating their insertion at their chain end by increasing their electrophilicity.

However, the effect of TEB and its amounts on the reactivity of both monomers is not well known. We thus carried a study to understand the effect of TEB and measure the variation of the monomer reactivity ratio and in particular that of EO in its presence. Starting from *r*_EO_ = 0 in absence of TEB, our objective was to precisely determine the reactivity ratio of EO for various content of TEB in the reaction medium. During the polymerization, the growing anion interact with TEB to form a weak nucleophile (a) as shown in Figure 1. This complex is able to ring-open EO activated by TEB to form alkoxide chain end (b). This chain end can then react with another TEB-activated EO (*k*_11_) to form an ether linkage (c) or with an anhydride (*k*_12_) to form ester linkage (d). The formed ester chain end (d) can only react with activated EO (*k*_21_) to form an alkoxide chain end (e), as the ring opening of another cyclic anhydride (*k*_22_) by a carboxylate ate complex to form a linear anhydride is not favored [19,20,21]. In absence of TEB, *k*_12_ >> *k*_11_ and *k*_21_ >> *k*_22,_
*k*_21_ are rate-determining steps. Here, we are interested in the determination of the effect of TEB loading on *r*_1_ = *k*_11_/*k*_12_ and on the reaction resulting in the formation of species (e).

### 2.1. Determination of Reactivity Ratio of EO as a Function of the Amount of TEB Used

Our initial investigation focused on the determination of reactivity ratios of ethylene oxide (*r*_EO_) for various amounts of TEB in the ROCOP of EO with anhydrides using in situ ^1^H NMR monitoring. With 1.3 eq. TEB for 1 eq. TBACl, 5 eq. SA and 100 eq. EO, it was observed that 52.8% of SA and 27.3% of EO were consumed in 2 h, and that 98% of SA and 88% of EO reacted in 5 h (entry 3, Table 1; Appendix A). Jaacks’ kinetic model was then used to calculate relative reactivity ratios of EO [18], and a value of *r*_EO_ = 0.52 was obtained for this experiment meaning that SA was relatively incorporated faster into polymer chains compared to EO irrespective of their initial concentrations.

The polymer chains after 2 h of reaction was obviously richer in EO units than in ester units, but it was richer in ester units compared to the polymer chains after 5 h. The growing chains thus undergo a drift in composition with the first part of the chains richer in ester functions and the last part richer in EO units. In the latter case, propagating alkoxide chain ends preferentially reacted with SA as compared to EO and *k*_12_ > *k*_11_. Increase in the amount of TEB to 1.7 eq. resulted in an increase of the reactivity of EO and a value of *r*_EO_ = 0.99 was observed; both SA and EO were thus incorporated into the polymer chains at the same rate, where *k*_12_ = *k*_11_ (entry 4, Table 1; Appendix A). This means that in the case of *r*_EO_ ≈ 1, there is no drift in composition as the copolymer chain grows; the copolymers formed are expected to retain the same composition as that of the initial monomer mixture throughout their growth. Increasing the amount of TEB to 2.3 eq. TEB further activated EO and a value of *r*_EO_ = 1.67 was obtained; in this case, EO was predominantly incorporated into the polymer chains, with *k*_12_ < *k*_11_ (entry 5, Table 1; Appendix A). The results indicate that the first equivalent of TEB form an ate complex with growing alkoxide and that the excess of TEB served to activate EO. From a situation where both SA and EO copolymerize in an alternating mode in the absence of TEB, one can see and conclude that the addition of the Lewis acid TEB selectively activated EO and helps to randomize the copolymer formed (entry 3 vs. entry 4 vs. entry 5, Table 1).

When phthalic anhydride (PA) was used as anhydride, 2.3 eq. of TEB was needed to achieve *r*_EO_ = 0.60, a higher amount of TEB compared to the reaction with SA presumably because of the competing interaction with TEB due to the presence of aromatic ring of PA (entry 6, Table 1; Appendix A); 43% PA and 21% EO were reacted in 0.5 h, and 97% PA and 88% EO conversions were achieved in 2 h. As expected, a further increase in the amount of TEB to 3.8 eq. increased the reactivity of EO and resulted in a value of *r*_EO_ = 1.1; 45% PA and 46% EO were reacted in 0.5 h, and 87% PA and 89% EO conversions were achieved in 2 h, wherein PA was almost equally distributed throughout the PEO polymer chain backbone (entry 7, Table 1; Appendix A). Further increase in amount of TEB to 4.1 eq. resulted in *r*_EO_ = 1.66; 52% PA and 62% EO were reacted in 0.5 h, and 80% PA and 92% EO conversions were achieved in 2 h (entry 8, Table 1; Appendix A). In the above reactions, for instance, after 0.5 h reaction time, 21%, 46%, and 62% conversions in EO and 43%, 45%, and 52% conversions in PA were achieved for 2.3 eq., 3.8 eq., and 4.1 eq. of TEB, respectively. It can be observed from these reactions that an increase in amount of TEB results in specific activation of EO. With 2.3 eq. and 4.1 eq. of TEB, one observes a drift of composition along the chain formed, and with 3.8 eq. of TEB no compositional drift is seen. Hence, TEB can be used as a tool for ROCOP of EO and anhydrides to synthesize polyether chains including degradable ester functions regularly distributed along the chains.

### 2.2. Synthesis and Characterization of Degradable PEOs

High-molar-mass PEO samples with degradable ester linkages were thus prepared under conditions varying with the *r*_EO_ values targeted. For all the reactions, 1 eq. TBACl, 5 eq. anhydride, 500 or 1000 eq. EO were used and only the amount of TEB was varied to obtain polymers with similar molar mass but different polymer microstructures due to different reactivity ratios of EO (*r*_EO_) (entries 1–7, Table 2). With 1.3 eq. TEB for SA/EO copolymerization, a value of *r*_EO_ = 0.52 was expected (entry 3, Table 1), resulting in a compositional drift along the growing chain (entry 1, Table 2). For 1.7 eq. TEB corresponding to a value of *r*_EO_ = 0.99 (entry 4, Table 1), SA and EO tend to uniformly distribute over the PEO chains (entry 2, Table 2). No compositional drift was expected in this case. In another experiment (entry 3, Table 2), the same amount of TEB loading and 500 eq. EO were used to produce short chains after chemical degradation (entry 3, Table 2). For 2.3 eq. TEB, which corresponds to a value of *r*_EO_ = 1.67 (entry 5, Table 1), EO can be expected to be further activated; the homopolymerization of EO dominating ester formation, 100% EO was consumed even before complete consumption of SA (entry 4, Table 2). Hence, a high-molar-mass PEO sequences were expected to be formed in this case. Similar set of reactions were performed with PA by varying TEB loadings from 2.3 eq. (entry 5, Table 2) to 3.8 eq. (entry 6, Table 2) and 4.1 eq. (entry 7, Table 2) to produce high molar mass PEO chains including phthalate units.

All polymer samples were chemically degraded by hydrolysis under basic conditions to further investigate the effect of TEB on the polymer microstructure, more specifically on the distribution of anhydride moieties along the PEO backbone. For the chemical degradation, 10 mg polymer samples were dissolved in water, and 0.5 M NaOH was added and allowed to undergo ester hydrolysis at 60 °C until complete degradation was achieved (followed by ^1^H NMR). An aliquot of the reaction mixture was neutralized with 0.1 M HCl, and the resultant PEO polyols were extracted into dichloromethane for further characterization. For sake of discussion, the degradation results were discussed in three categories depending on the values of *r*_EO_ whether (i) *r*_EO_ < 1 or (ii) *r*_EO_ = 1 or (iii) *r*_EO_ > 1.

(i)Reactions with *r*_EO_ < 1 (entries 1 and 5 in Table 2):

For a SA/EO copolymerization comprising 1.3 eq. of TEB (*r*_EO_ = 0.52), the polymer formed exhibited a molar mass of 36.1 kg/mol and after degradation a molar mass of 8.21 kg/mol (entry 1, Table 2; Appendix A). Similarly, for a PA/EO copolymerization reaction comprising 2.3 eq. of TEB (*r*_EO_ = 0.60), the polymer formed showed a molar mass of 36.2 kg/mol and upon degradation 8.66 kg/mol as molar mass (entry 5, Table 2; Appendix A). In both cases, all initial anhydrides were incorporated compared to only 88% of EO consumed affording comparatively short PEO chains after degradation. Because of the compositional drift along the generated PEO chains, the PEO diols obtained after degradation exhibited broad polymer chain distribution.

(ii)Reactions with rEO = 0.99 to 1.1 (entries 2, 3, and 6 in Table 2):

For SA/EO copolymerization reactions carried out with 1.7 eq. of TEB (*r*_EO_ = 0.99), the polymer produced exhibited a molar mass of 38.0 kg/mol and after degradation the sample isolated showed 8.34 kg/mol as molar mass of PEO diol (entry 2, Table 2; Appendix A); for the P(EO-*co*-ester) sample of molar mass 21.6 kg/mol, its degradation produced a PEO diol sample of 4.03 kg/mol molar mass (entry 3, Table 2). Similarly, for PA/EO copolymerization comprising 3.8 eq. of TEB (*r*_EO_ = 1.1), a P(EO-*co*-ester) of 43.2 kg/mol; its degradation produced a PEO diol of a molar mass 8.97 kg/mol (entry 6, Table 2; Appendix A). As the reactivity ratios of EO for these reactions were close to unity, it is expected that anhydrides were uniformly distributed over the PEO chains; the degradation results showed a rather narrow chain distributions of all PEO diols obtained (entries 2, 3, and 6 in Table 2).

(iii)Reactions with rEO > 1 (entries 4 and 7 in Table 2):

For a SA/EO copolymerization with 2.3 eq. of TEB (*r*_EO_ = 1.67), the P(EO-*co*-ester) sample obtained showed a molar mass of 39.6 kg/mol; its degradation generated a PEO diol of 9.35 kg/mol molar mass (entry 4, Table 2; Appendix A). Similarly, for a PA/EO copolymerization carried out with 4.1 eq. of TEB (*r*_EO_ = 1.66), a sample of 43.9 kg/mol molar mass was obtained that afforded PEO diol of 10.4 kg/mol (entry 7, Table 2; Appendix A). As observed from in situ kinetic reactions data, EO is more activated by the excess of TEB; it is therefore completely consumed even before complete conversions of either SA or PA. Hence, comparatively longer PEO chains and broader chain distribution of resulting PEO diols were observed after degradation.

## 3. Materials and Methods

### 3.1. Materials and Methods

Succinic anhydride and phthalic anhydride were purchased from Sigma-Aldrich and purified according to the literature procedure prior to use [22]. In a modified procedure, phthalic anhydride was recrystallized in chloroform (10 g/100 mL), sublimed three times, and dried over P_2_O_5_ under argon immediately prior to use. 1M TEB in THF solution was purchased from Aldrich and Acros and used as received. Tetrabutylammonium chloride (TBACl) was purchased from Sigma-Aldrich and dried over P_2_O_5_ in argon environment prior to use. ^1^H NMR spectra were recorded on a Bruker AVANCE III-400/500 Hz instrument in THF-*d*_8_ or CDCl_3_. GPC traces were recorded by Agilent 1260 Infinity equipped with Agilent guard column PLgel 5 um 50 × 7.5 mm and two columns of PLgel 10 um MIXED-B 300 × 7.5 mm using DMF (1 mL/min) as the eluent at 40 °C and 0.1% lithium chloride salt is added to the mobile phase DMF. Polyethylene glycol/polyethylene oxide standards are used as reference standards to establish a calibration curve.

### 3.2. Representative Procedure for Copolymerization of EO and Anhydride for In Situ Kinetic NMR Experiment

Inside a glovebox, 1 eq. TBACl, the required amount of 1M TEB in THF, and 5 eq. anhydride were added in same order into a dry J-Young NMR tube, and the mixture is dissolved in 0.6 mL THF-*d*_8_. A cooled syringe was used to transfer 100 eq. EO from a cooled EO containing Schlenk flask to the above solution in the NMR tube and sealed immediately. In less than a minute, the solution in the NMR tube was freeze in liquid nitrogen to cease any reaction to proceed. Once the solution has attained near to room temperature, a kinetic NMR program was run to monitor reaction progress where a spectrum was collected every 5 min until completion of the reaction. After complete conversion was achieved, the reaction mixture was quenched with 0.5 M HCl in THF and precipitated in diethyl ether. The purified product was dried in vacuum at room temperature and characterized by ^1^H NMR and GPC.

### 3.3. In Situ Kinetic NMR Experiment Data Processing and Calculation of Reactivity Ratios by Jaacks Model

MestReNova NMR software version 11.0.1 was used to process the raw data. All the individual NMR spectra were baseline corrected and their peak positions were adjusted using solvent residual signals from THF-*d*8 at 1.72 ppm and 3.58 ppm as reference. A sextet at 1.37 ppm in the ^1^H NMR representing 8 Hs from an alkyl group of tetrabutyl ammonium cation part of the initiator was calibrated to 8 in all the kinetic NMR spectra. All the remaining peaks were integrated with reference to the 1.37 ppm peak. Using these data, a graph was plotted for log[M_AH_/M_AHo_] vs. log[M_EO_/M_EOo_] to determine reactivity ratio of ethylene oxide (*r*_EO_) for Jaacks kinetic model. From the *y* = *mx* plot, slope *m* is *r*_EO_. [M_AH_] and [M_EO_] are concentrations of unreacted anhydride and EO, respectively, at respective time point as obtained from ^1^H NMR, and [M_AHo_] and [M_Eoo_] are initial concentrations of both monomers.

### 3.4. Representative Procedure for Copolymerization of EO and Anhydride for Schlenk Tube Experiment

Inside a glovebox, 1 eq. TBACl, the required amount of 1M TEB in THF, and 5 eq. anhydride were added in the same order into a dry Schlenk flask equipped with a magnetic stirrer and dissolved in 5 mL dry THF. A cooled syringe was used to transfer required amount of EO from a cooled EO containing Schlenk flask to the above solution and sealed immediately. The flask was stirred at room temperature until solution turned viscous. After complete conversion was achieved, the reaction mixture was quenched with 0.5 M HCl in THF and precipitated in diethyl ether. The purified product was dried in vacuum at room temperature and characterized by ^1^H NMR and GPC.

### 3.5. Representative Procedure for Degradation of PEO Based Poly(ester Ether) Copolymers

10 mg of a pure polymer is dissolved in 2 mL deionized water that resulted in a colorless solution. 3 mL 0.5 M NaOH solution is added to the above solution and stirred at 60 °C until complete degradation is achieved. An aliquot of the above solution is neutralized with 10% HCl in H_2_O and the degraded polymer is extracted from solution by dichloromethane. The resultant product is dried under vacuum and characterized by ^1^H NMR and GPC.

## 4. Conclusions

The copolymerization of epoxides, namely of EO, with anhydrides is known to produce perfectly alternating copolymers, reflecting reactivity ratios close to 0 for both types of monomers. In the presence of TEB, we demonstrate in this investigation that the reactivity ratio of EO can be progressively raised. The first equivalent of TEB forms an ate complex with the growing anionic species, preventing the occurrence of side reactions such as back-biting reactions and transesterification reactions. The excess amount of TEB served to activate the monomers. It is shown in this study that the excess of TEB mainly activates EO rather than anhydrides. Depending upon the amount of TEB used, the values of *r*_EO_ could be varied at will from values below 1 to values higher than 1.

Conditions were even found to carry out copolymerizations of EO with AH with *r*_EO_ equal or close to 1. In the latter case, no compositional drift was observed in the P(EO-*co*-ester) chains formed, which is reflected in the narrow distribution of molar masses of PEO diols isolated after basic degradation. For P(EO-*co*-ester) samples produced under values of *r*_EO_ < 1 or *r*_EO_ > 1, the compositional drift observed resulted in PEO diol chains of broader molar mass distribution after degradation.

## Data Availability

Data is contained within the article or Appendix A.

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
