# Peer review of "Using Triethylborane to Manipulate Reactivity Ratios in Epoxide-Anhydride Copolymerization: Application to the Synthesis of Polyethers with Degradable Ester Functions"

_molecules, 2022, doi:10.3390/molecules27020466_

Round 1
Reviewer 1 Report
Overall, the work presented in this paper is interesting because of the reliable analysis. This manuscript was recommended to be accepted as it is.
Author Response
Thanks a lot for comments!
Reviewer 2 Report
In this submitted manuscript, Dr. Gnanou, Dr. Feng, and Vamshi Chidara studied an approach to manipulate the reactivity ratios of epoxides and anhydrides during an anionic ring-opening copolymerization. Following that, the degradation of ester functional groups was studied under mild basic conditions. It is interesting to find that in the presence of triethylborane (TEB), the reactivity ratio of ethylene oxide (EO) could increase from 0 to a value lower or higher than 1 at will. For the copolymer, P(EO-co-ester), which has minimal compositional drift, affords PEO diol samples of narrow molar mass distribution after degradation under basic conditions.
This is a nice follow-up work with the authors’ previous research on studying the role of TEB in the preparation of epoxide-based copolymers via anionic ring-opening copolymerization (Polym. Chem. 2019, 10, 3764-3771; J. Am. Chem. Soc. 2016, 138, 11117-11120; Macromolecules 2021, 54, 2711-2719), and investigating on the manipulation of EO reactivity ratio is challenging and important, however, the manuscript in its current form has evident weaknesses and the work falls short in terms of rigor and care. In many aspects, the authors should give more thorough and reasonable explanations, and the manuscript needs careful proofreading as the data in Table 2 is not correct. Based on the content and importance of this manuscript, I would recommend rejection and resubmission for further considerations.
Criticisms include:
- In Table 2 on page 6, most of the data is not correct, as the authors messed it up in the table. Except for columns of the ratio for monomers, initiator, and base, and the Mn / PDI from GPC after degradation, all other columns were messed up. Actually, in the entry 1 and 2, the molar ratios of monomers, initiator, and base were also wrong. This is a very evident error, and because of this, the sections of 2.2 (i) – 2.2 (iii) were not reviewed as the data was not correct.
- It shows conflicting results from previous studies and the observations reported in this manuscript. From lines 64-65, Dr. Zhang and co-authors reported that in the presence of TEB, the alternating copolymerization of EO with anhydride automatically shifted to homopolymerization of EO. And in lines 70-71, the authors claimed that EO and anhydrides copolymerized statistically, resulting in a random distribution of ester functions along the polyether chain. It is confusing that in the presence of TEB, what type of (co)polymers could be formed on earth, homopolymer (PEO) or statistical random copolymer P(EO-co-anhydride)??
- In Figure 1, the topology of copolymers should be determined based on the reactivity ratios of monomers, not just rEO. In the later section (Scheme 1) the authors mentioned that the reactivity ratio of anhydride is 0, however, this should be clearly clarified in Figure 1, otherwise, the information to draw the topology of copolymers is not completed. Another question for Figure 1 is why it forms “ester rich” and “ester evenly distributed” copolymers when rEO <1 and rEO = 1, respectively?
- In Scheme 1 the authors draw the structure of ate complex b, however, is that a correct intermediate in an anionic ring-opening polymerization? Is the triethylborane unit directly connected with oxygen from the epoxide? In a normal anionic ring-opening polymerization, the nucleophile (TEB) should covalently connect with a carbon atom, and form an oxygen anion (alkoxide) as the new nucleophile to attack another epoxide.
- The authors should explain why the intermediate “d” in Scheme 1 could not react with another anhydride (k22 = 0), but can ring open another epoxide (lines 122-123).
- In the paragraph from lines 146 to 163, the authors used the word “latter” many times, and in some sentences, it is hard to understand what is the latter designated to. It should be revised to make the designation more clear.
- Line 166, the authors should explain why in the presence of aromatic rings, the required amount of TEB is increased in order to reach a similar reactivity ratio of EO compared with the cases using SA as the anhydrides.
- Lines 177-178, what would be the reasonable explanation for the observation that 2.3 eq and 4.1 eq of TEB give a drift of composition, while 3.8 eq of TEB shows no compositional drift.
- Rewrite the whole section 2.2, as the data was not correct and didn’t match from paragraphs to Table 2.
- Line 206, what is the technique or method used to determine if the complete degradation was achieved or not?
- Line 255, the authors mentioned that the phthalic anhydride was sublimed three times after recrystallization, is that true? Sublimation requires a high temperature and is the compound stable enough during the sublimation process, considering it goes through three times?
- Some sentences need to be revised or rewritten, for example, line 77, “As practical…” should be revised as “As a practical…”; Scheme 1 the letter “c” is in different color and not bold; Line 254, the literature for purifying anhydrides should be cited.
Author Response
Comments from Reviewer-2
Comments and Suggestions for Authors
In this submitted manuscript, Dr. Gnanou, Dr. Feng, and Vamshi Chidara studied an approach to manipulate the reactivity ratios of epoxides and anhydrides during an anionic ring-opening copolymerization. Following that, the degradation of ester functional groups was studied under mild basic conditions. It is interesting to find that in the presence of triethylborane (TEB), the reactivity ratio of ethylene oxide (EO) could increase from 0 to a value lower or higher than 1 at will. For the copolymer, P(EO-co-ester), which has minimal compositional drift, affords PEO diol samples of narrow molar mass distribution after degradation under basic conditions.
This is a nice follow-up work with the authors’ previous research on studying the role of TEB in the preparation of epoxide-based copolymers via anionic ring-opening copolymerization (Polym. Chem. 2019, 10, 3764-3771; J. Am. Chem. Soc. 2016, 138, 11117-11120; Macromolecules 2021, 54, 2711-2719), and investigating on the manipulation of EO reactivity ratio is challenging and important, however, the manuscript in its current form has evident weaknesses and the work falls short in terms of rigor and care. In many aspects, the authors should give more thorough and reasonable explanations, and the manuscript needs careful proofreading as the data in Table 2 is not correct. Based on the content and importance of this manuscript, I would recommend rejection and resubmission for further considerations.
Criticisms include:
- In Table 2 on page 6, most of the data is not correct, as the authors messed it up in the table. Except for columns of the ratio for monomers, initiator, and base, and the Mn / PDI from GPC after degradation, all other columns were messed up. Actually, in the entry 1 and 2, the molar ratios of monomers, initiator, and base were also wrong. This is a very evident error, and because of this, the sections of 2.2 (i) – 2.2 (iii) were not reviewed as the data was not correct.
Answer: we indeed made a major mistake upon submitting our initial manuscript. Upon receiving the comments from the reviewer, we revised the manuscript with a corrected version of Table 2 which has been uploaded to replace the initial version.
2. It shows conflicting results from previous studies and the observations reported in this manuscript. From lines 64-65, Dr. Zhang and co-authors reported that in the presence of TEB, the alternating copolymerization of EO with anhydride automatically shifted to homopolymerization of EO. And in lines 70-71, the authors claimed that EO and anhydrides copolymerized statistically, resulting in a random distribution of ester functions along the polyether chain. It is confusing that in the presence of TEB, what type of (co)polymers could be formed on earth, homopolymer (PEO) or statistical random copolymer P(EO-co-anhydride)??
Answer: As demonstrated by our work, the amount of TEB significantly affects the reactivity of EO during copolymerization. In the case of the work reported by Dr. Zhang and co-authors, a very low amount of TEB with respect to initiator (t-BuP1/Et3B = 0.5/0.15) was used, leading to the formation of strictly alternating polyester (rEO = 0) first and of a second block of PEO due to the homopolymerization of remaining EO after complete consumption of anhydride.
In the present manuscript, we discuss three different categories namely rEO<1, rEO=1, and rEO>1 as a function of TEB content.
- In the presence of a low content in TEB content (entries 3, 5 in Table 1, entries 1, 5 in Table 2) leading to rEO<1, ester preferentially forms over ether until anhydride is completely consumed. Initial parts of polymer chains will thus be ester-rich, shifting progressively to ether-rich ending parts of the chains.
- In presence of the optimal amount of TEB content (entries 4, 7 in Table 1, entries 2, 3 and 6 in Table 2) leading to rEO ≈1, statistical random copolymer P(EO-co-anhydride) can be formed.
- In presence of high amount of TEB content (entries 5, 8 in Table 1, entries 4, 7 in Table 2) leading to rEO>1, ether preferentially forms over ester in the polymer chain.
3. In Figure 1, the topology of copolymers should be determined based on the reactivity ratios of monomers, not just rEO. In the later section (Scheme 1) the authors mentioned that the reactivity ratio of anhydride is 0, however, this should be clearly clarified in Figure 1, otherwise, the information to draw the topology of copolymers is not completed. Another question for Figure 1 is why it forms “ester rich” and “ester evenly distributed” copolymers when rEO <1 and rEO = 1, respectively?
Answer: it is known that cyclic anhydrides cannot homopolymerize, meaning that their respective reactivity ratio is equal to zero. As shown in structure (d) of scheme1 anhydride cannot react with another anhydride, k22=0. Therefore, this value was not added in Figure 1.
As for the meaning of “ester-rich” and “ester evenly distributed” copolymers, it indicates that the distribution of esters along the formed polymer chain can amply vary as thoroughly discussed in the last paragraph of Results and Discussion (L220-250).
4. In Scheme 1 the authors draw the structure of ate complex b, however, is that a correct intermediate in an anionic ring-opening polymerization? Is the triethylborane unit directly connected with oxygen from the epoxide? In a normal anionic ring-opening polymerization, the nucleophile (TEB) should covalently connect with a carbon atom, and form an oxygen anion (alkoxide) as the new nucleophile to attack another epoxide.
Answer: the formation of ate complex B has already been well confirmed by 1H and 11B NMR characterization by us and colleagues of other research groups in published papers. In the case of epoxide, the oxygen atom is more basic than carbon atom to interact with TEB.
5. The authors should explain why the intermediate “d” in Scheme 1 could not react with another anhydride (k22 = 0), but can ring open another epoxide (lines 122-123).
Answer: Under present reaction conditions the ring-opening of another cyclic anhydride (k22) by a carboxylate to form a linear anhydride is not favored in the presence of activated epoxide monomer. Such results were observed in similar published works of copolymerization of epoxides and cyclic anhydrides (Green Chem. 2019, 21, 6123-6132, Polym. Chem. 2018, 9, 4052-4062; Macromolecules 2018, 51, 3126-3134). The text in L 122-123 was modified.
6. In the paragraph from lines 146 to 163, the authors used the word “latter” many times, and in some sentences, it is hard to understand what is the latter designated to. It should be revised to make the designation more clear.
Answer: this paragraph was modified to make it clearer as suggested by the reviewer.
7. Line 166, the authors should explain why in the presence of aromatic rings, the required amount of TEB is increased in order to reach a similar reactivity ratio of EO compared with the cases using SA as the anhydrides.
Answer: One explanation might be that the aromatic ring of PA competes with epoxide to interact with TEB, thereby demanding more amount of TEB. The sentence in Line 166 was modified accordingly.
8. Lines 177-178, what would be the reasonable explanation for the observation that 2.3 eq and 4.1 eq of TEB give a drift of composition, while 3.8 eq of TEB shows no compositional drift.
Answer: with different amount TEB added, different reactivity ratio rEO values were obtained, affording copolymers with different structures, which have been already discussed in answers to questions 2 and 3.
9. Rewrite the whole section 2.2, as the data was not correct and didn’t match from paragraphs to Table 2.
Answer: Table 2 was already corrected in updated version and now matches the text of manuscript.
10. Line 206, what is the technique or method used to determine if the complete degradation was achieved or not?
Answer: Complete degradation was confirmed by 1H NMR where the peak corresponding to ester at 4.35 ppm completely disappeared.
11. Line 255, the authors mentioned that the phthalic anhydride was sublimed three times after recrystallization, is that true? Sublimation requires a high temperature and is the compound stable enough during the sublimation process, considering it goes through three times?
Answer: Yes, Phthalic anhydride is first recrystallized from chloroform and submitted to sublimation 2-3 times under reduced pressure at 80 ºC instead of only at high temperature. The sublimed anhydride was stable and pure as confirmed from 1H NMR spectroscopy. The reference was cited.
12. Some sentences need to be revised or rewritten, for example, line 77, “As practical…” should be revised as “As a practical…”; Scheme 1 the letter “c” is in different color and not bold; Line 254, the literature for purifying anhydrides should be cited.
Answer: corrected, and reference was cited (J. Am. Chem. Soc. 2016, 138, 7107-7113).

Reviewer 3 Report
The paper can lead to the development of novel polymeric compounds. The paper is interesting and well-presented. The experimental design is sound and the conclusions are supported by the data. I believe that the paper is suitable for publication in this journal.
Author Response
Thanks for your comments!
Round 2
Reviewer 2 Report
The authors have made a thorough revision for the manuscript titled "Using Triethylborane to Manipulate Reactivity Ratios in Epoxide-Anhydride Copolymerization: Application to the Synthesis of Polyethers with Degradable Ester Functions". The questions and concerns from the reviewer have been solved and explained.
The manuscript has been greatly improved in its quality and significance, and I would recommend accepting it in its present form.